# Research and Challenges of Reinforcement Learning in Cyber Defense Decision-Making for Intranet Security

**Wenhao Wang** [1] , **Dingyuanhao Sun** [2], **Feng Jiang** [2], **Xingguo Chen** [2] **and Cheng Zhu** [1,*]

1   Science and Technology on Information Systems Engineering Laboratory, National University of Defense Technology, Changsha 410073, China; wangwenhao11@nudt.edu.cn
2   Jiangsu Key Laboratory of Big Data Security & Intelligent Processing, Nanjing University of Posts and Telecommunications, Nanjing 210023, China; 1021041203@njupt.edu.cn (D.S.); 1320047710@njupt.edu.cn (F.J.); chenxg@njupt.edu.cn (X.C.)
*   Correspondence: zhucheng@nudt.edu.cn

**Abstract:** In recent years, cyber attacks have shown diversified, purposeful, and organized characteristics, which pose significant challenges to cyber defense decision-making on internal networks. Due to the continuous confrontation between attackers and defenders, only using data-based statistical or supervised learning methods cannot cope with increasingly severe security threats. It is urgent to rethink network defense from the perspective of decision-making, and prepare for every possible situation. Reinforcement learning has made great breakthroughs in addressing complicated decision-making problems. We propose a framework that defines four modules based on the life cycle of threats: pentest, design, response, recovery. Our aims are to clarify the problem boundary of network defense decision-making problems, to study the problem characteristics in different contexts, to compare the strengths and weaknesses of existing research, and to identify promising challenges for future work. Our work provides a systematic view for understanding and solving decision-making problems in the application of reinforcement learning to cyber defense.

**Keywords:** reinforcement learning; intelligent decision-making model; cyber defense; decision-making framework



## 1. Introduction

The openness and security in cyberspace have always been conflicting issues. Enterprises, governments and schools hope to provide convenient services. At the same time, nobody wants their confidential data stored and the key systems in the internal network to be controlled by malicious cyber attackers. On 26 May 2021, the National Computer Network Emergency Response Technical Team of China (CNCERT) pointed out that multiple attacks continued to increase during the Coronavirus Disease 2019 (COVID-19) pandemic [1]. As the most threatening form of attack to large organizations or enterprises, Advanced Persistent Threat (APT) attacks compromised the mail servers of some local government departments by sending phishing emails related to COVID-19, which want to obtain more classified intelligence in the internal network. Apart from APT, the ransomware attacks are on the rise in 2020. A large number of corporate networks have been attacked, resulting in serious economic losses. From the analysis of ransomware, it can be seen that technical means of ransomware are constantly escalating and the selection of targets is becoming smarter. The above trends all indicate that cyber attacks on intranet security are becoming more targeted and organized.

(1)   Targeted: On one hand, the development of attack techniques has enriched the existing attack surface. Experienced attackers who are able to integrate collected information and knowledge to choose appropriate actions in a task-oriented manner. Therefore, isolated defense technology no longer works in these cases, security experts need to make decisions under different threat situations, which requires more precise objectives.

(2) Organized: In order to obtain the continuous attack effect, the attacker will constantly look for vulnerabilities, and carry out more intelligent attacks by coordinating the attack resources. To fight against the attacker, it is also necessary for the defender to allocate defense resources to deal with both known and unknown threats.

These changes make intranet security increasingly challenging. Due to the continuous upgrading of attack techniques and tools, the traditional data mining based on data, machine learning, deep learning or statistical methods [2–6] cannot solve the problem of how to adapt to changes of cyber attacks, which require a new approach from the perspective of decision-making. Reinforcement learning (RL) algorithms are a popular paradigm for solving decision-making problems under complex interactions. Combined with the powerful expression mechanism of deep learning, it can effectively solve decision-making problems in a large state space. It has been successfully applied to games [7–9], chess [10,11], robots [12,13] and other fields, showing its advantages to help human decision-making. Figure 1 shows the number of relevant literature from 2016 to 2021. It can be seen that applications of reinforcement learning to cyber decision-making has drawn much attention from academia.

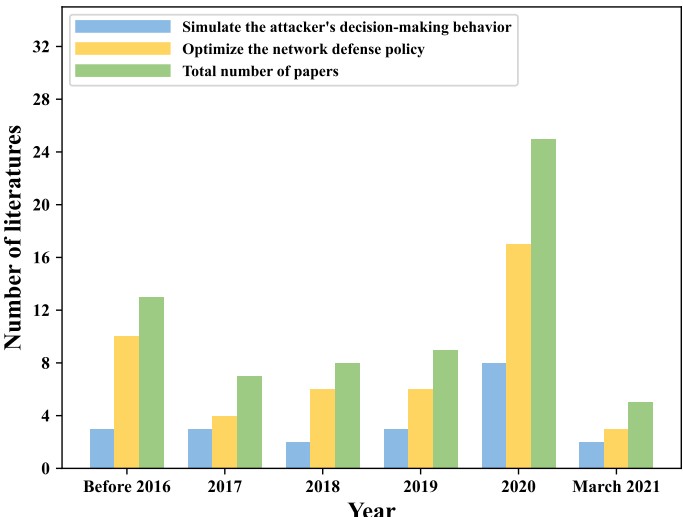

**Figure 1.** Research trend of cyber defense decision-making based on reinforcement learning.

However, how reinforcement learning can be used to address issues in cyber defense decision-making remains unclear. The researchers group applications of reinforcement learning in cybersecurity into DRL-based security methods for cyber-physical systems, autonomous intrusion detection techniques, and multiagent DRL-based game theory simulations [14], but do not specify how these different classifications are used to support cyber defense decision-making tasks. We extend this perspective, and focus on cyber defense decision-making for the internal network from both the attacker's and defender's perspective. The problem is analyzed from three aspects: problem characteristics, decision-making models and reinforcement learning algorithms. The characteristics of the problem are the basis of decision-making models and reinforcement learning algorithms, and determine the specific decision-making models to be used. Based on the selected models, reinforcement learning algorithms are applied to solve the problem. The major contents presented in this study are shown in Figure 2. In this survey, we make the following contributions:

(1) A new framework PDRR (Pentest Design Response Recovery) was proposed to distinguish different cyber defense decision-making tasks. The framework aimed to demarcate the boundaries of the problems, which can help researchers better understand the cyber defense decision-making problem;

(2) Based on the proposed framework, the existing research literature was summarized from the perspectives of the attack and defender respectively. We first discussed the

problem characteristics from different perspectives, and then categorized literature according to the type of problem and the adopted decision-making model, and finally compared the problem characteristics, strategies, tasks, advantages and disadvantages used in different literature.

(3) After summarizing the existing literature, we analyzed the future direction and challenges from the perspectives of reinforcement learning algorithms and cyber defense decision-making tasks respectively. The purpose of this discussion is to help researchers understand the cyber defense decision-making task more clearly, and to promote the applications of reinforcement learning algorithms to this task.

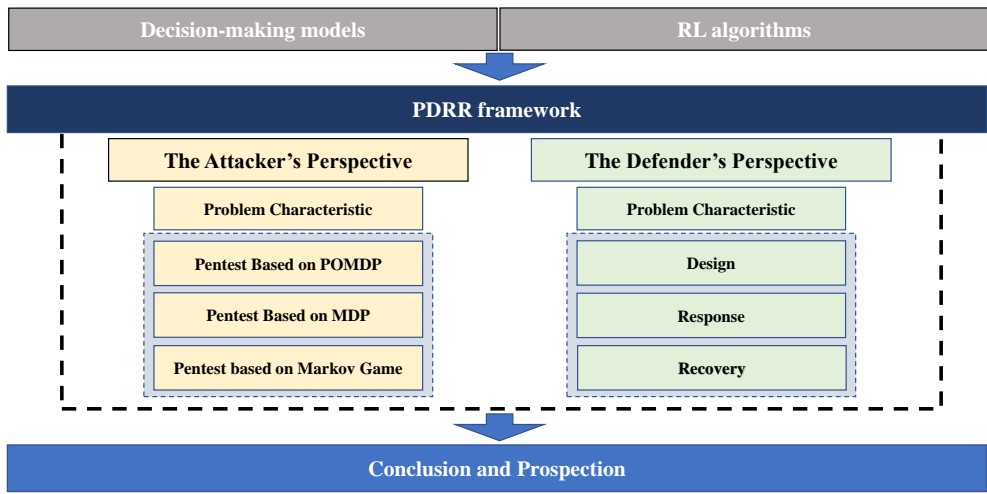

**Figure 2.** The structure of this paper.

The rest of this paper is organized as follows. First, we introduce the research background in Section 2. We present a framework of network defense decision-making in Section 3. In Sections 4 and 5, we summarize and compare the existing reinforcement learning research on network defense decision-making. After a summary of current works, the future research directions and challenges are pointed out in Section 6. Finally, Section 7 is the conclusion of the whole paper.

## 2. Research Background

In this section, we provide a brief overview of decision-making models and RL algorithms. The decision-making model abstracts interactions between decision makers and the environment, and the RL algorithm looks for optimal policies under different problem scenarios.

### 2.1. Decision-Making Models

In a decision-making task, an agent perceives states (or observations) from the environment and takes an action. The environment transfers to a new state (or observation) and feeds back to the agent for a reward. This interaction process between the agent and the environment will keep going or end when it reaches the end state. In this process, the agent learns an optimal policy to maximize the accumulated reward. The decision-making model has a variety of formal abstractions, in which the most classic one is the Markov Decision Process (MDP) [15]. Beyond that, the most commonly used intelligent decision-making models for cyber defense decision-making tasks include (1) Markov Game which is extended from one agent to multiple agents [16]; (2) a Partially Observable Markov Decision Process (POMDP) [17–19] which is obtained from a fully observable state to a partially observable state.

MDP: Only one agent exists in a decision-making task, and the environment satisfies the Markov property, which means that the change of state depends only on the current state and action, instead of the historical state and action. An MDP involves a quadruple

$\langle S, A, R, T \rangle$, where $S$ indicates the status space, and $A$ indicates the executable action space $R : s, a, s' \in S \times A \times S \to r = R(s, a, s') \in \mathbb{R}$ is a bounded reward function, and $T : S \times A \times S \to [0, 1]$ is a probability function of state transition, which satisfies the $\forall s, a$, $\sum_{s' \in S} T(s, a, s') = 1$. With its interacting with the Environment, the Agent will learn and optimize a policy $\pi : S \times A \to [0, 1]$, which is made to maximize the cumulative rewards over the long term $Return = \mathbb{E}_\pi \left[ \sum_{t=0}^\infty \gamma^t r_t \right]$, where $\gamma \in (0, 1]$ indicates the discount rate, $t$ indicates the discrete time step, and $r_t = R(s_t, a_t, s_{t+1})$ indicates the Rewards for $t$ time, $a_t \sim \pi(s_t, a_t)$.

Markov Game: It is a quintuple consisting of agents:$\langle N, S, \{A_i\}_{i=1}^n, \{R_i\}_{i=1}^n, T \rangle$. Among them, $N$ represent the set of agents, $n$ represent the quantity of agents, and $A_i$ represents the action space of the agent $i$. If $A = A_1 \times A_2 \cdots \times A_n$ is the joint action space for all agents, then $R_i : S \times A \times S \to \mathbb{R}$ is the reward function for the Agent $i$. $R_i(s, \vec{a}, s')$ indicates the reward for Agent $i$ fed back by the Environment after all agents take a joint action $\vec{a}$ on the state $S$, and $T(s, \vec{a}, s')$ represents the probability that the Environment moves from the state $s$ to the subsequent state $s'$ through the joint action $\vec{a}$.

POMDP: In this process, an Agent cannot obtain the true state of the Environment, but can only observe a part of the state or the state after interference. This is called observation. POMDP is a six-tuple $\langle S, A, R, T, \Omega, O \rangle$, where $\langle S, A, R, T \rangle$ constitute a potential MDP, $\Omega$ indicates the limited set of Observations available to an Agent, and the Observation function $O : S \times A \times \Omega \to [0, 1]$ indicates that the Observation $o$ is obtained according to the Probability $O(o|s', a)$ on the subsequent state $s'$ after the action $a$ is adopted on the state $s$, and the reward $R(s, a, s')$ is also obtained.

In addition, according to the time of the Agent's decision-making from discrete time step to continuous retention time, the Semi-Markov Decision Process (SMDP) [20,21], Continuous-time Markov Decision Process (CTMDP) [22–24] and other decision-making models can also be applied.

*2.2. RL Algorithms*

The basic decision-making model of RL is MDP. When the state and action space is small, RL assigns a value to each state or state-action pair, where $V^\pi(s)$ denotes the value of state $s$.

$$V^\pi(s) \doteq \mathbb{E}_\pi \left[ \sum_{t=0}^\infty \gamma^t r_t | s_0 = s \right], \tag{1}$$

$Q^\pi(s, a)$ denotes the value of state-action pair $(s, a)$.

$$Q^\pi(s, a) \doteq \mathbb{E}_\pi \left[ \sum_{t=0}^\infty \gamma^t r_t | s_0 = s, a_0 = a \right]. \tag{2}$$

By definition, the Bellman equation is obtained:

$$V^\pi(s) = \mathbb{E}_\pi \left[ R(s, a, s') + \gamma V^\pi(s') \right]. \tag{3}$$

Taking the state value function as an example, the Bellman equation can be defined by Equation (4).

$$\begin{aligned} V^\pi(s) &= \mathbb{E}_\pi \left[ \sum_{t=0}^\infty \gamma^t r_t | s_0 = s \right] \\ &= \mathbb{E}_\pi \left[ R(s, a, s') + \gamma \sum_{t=0}^\infty \gamma^t r_t | s_0 = s \right] \\ &= \mathbb{E}_\pi \left[ R(s, a, s') + \gamma V^\pi(s') \right]. \end{aligned} \tag{4}$$

The optimal strategy $\pi^*$ satisfies $\forall s \in S$, $V^*(s) = \max_\pi V^\pi(s)$, where $V^*$ is the optimal value function, which satisfies Bellman's optimal equation.

$$V^*(s) = \max_a \mathbb{E}\left[ R(s, a, s') + \gamma V^*(s') \right].\tag{5}$$

Define the Bellman optimal operator $\mathbb{T} : \mathbb{R}^{|S|} \to \mathbb{R}^{|S|}$.

$$\mathbb{T}V(s) \doteq \max_a \mathbb{E}\left[ R(s, a, s') + \gamma V(s') \right].\tag{6}$$

The key step of the value iteration algorithm can be obtained: $V_{n+1} = \mathbb{T}V_n$. Synthesizing the multi-step Bellman evaluation operator $\mathbb{T}^\pi$, the operator $\mathbb{T}^{\lambda,\pi}$ that satisfies the compression map can be obtained in $\mathbb{T}^{\lambda,\pi}V \doteq (1-\lambda)\sum_{i=0}^{\infty} \lambda^i (\mathbb{T}^\pi)^{i+1}V$, and thus get the $\lambda$-policy iteration algorithm.

Therefore, policies can be expressed based on value functions, or can be defined directly. We summarize related RL algorithms based on tabular values, value function evaluation, and policy evaluation.

RL Algorithms Based on Tabular Values: Policy Iteration and its improved algorithm continue to repeat the two stages of Policy Evaluation and Policy Improvement until the optimal Policy and the optimal value function [25] is converged and obtained. However, the original Policy Iteration algorithm requires the value function to converge to the optimal solution for each Policy Evaluation before making a Policy Improvement. In the process of Policy Evaluation, the Value Iteration algorithm performs only one time of Iteration. When the state space is large, it takes a long time to scan all states. In this case, Asynchronous Value Iteration improves efficiency by iterating only one sample of the state space at a time. In the stage of Policy Evaluation, an optimal solution can be approximated more quickly with $\lambda$-Policy Iteration synthesizing all the multi-step Bellman evaluation operators and doing only one iteration; While Q-learning uses Bellman's optimal operator, which is $\max_a \mathbb{E}[R(s, a, s') + \gamma V(s')] \approx \max_a [R(s, a, s') + \gamma V(s')]$, to approximate the optimal value function and perform learning control. This Policy is characterized by the difference between behavioral policy and learning policy, falling into the off-policy learning.

RL Algorithms Based on Value Function Evaluation: Large-scale or continuous states and actions caused the curse of dimensionality, making the tabular value based RL algorithms ineffective. However, the ways to solve these this curse include: (i) to reduce the state and action space; (ii) to estimate the value function $V$ or Policy by using parameters far smaller than the number of states and actions. The state and action space is mapped by function to a set of parameter dimensions that are much smaller than the size of the state and action space. The models of function estimation are usually divided into linear model, kernel method, decision-making tree, neural network, deep neural network etc. DQN [7] is a Q-learning method using deep neural network as value function estimator. By correcting the over-estimation in Q-learning (such as Double Q-learning) [26], Double-DQN [27], Averaged-DQN [28], MellowMaxDQN [29] based on the MellowMax operator [30], Soft-MaxDQN [31] based on the SoftMax algorithm, and the soft greedy method based on Rankmax [32] were proposed successively.

RL Algorithms Based on Policy Evaluation: Unlike the Policy based on the representation of value function, RL based on Policy Evaluation explicitly defines the parameterized policy $\pi(a|s, \theta)$, and constantly optimizes the parameter $\theta$ by using the policy gradient ascent method according to the policy evaluation function for the parameter $J(\theta)$. The gradient of the policy is $\nabla J(\theta) \propto \sum_s \mu(s) \sum_a q_\pi(s, a) \nabla \pi(a|s, \theta)$. By using the random gradient ascent method and replacing $q_\pi(s, a)$ with Monte Carlo evaluation, the REINFORCE algorithm is obtained [33]. I the value function of state is updated by the Bootstrap method, the Actor-Critic algorithm is obtained [34]. To solve the continuous action space problem, the stochastic policy gradient algorithm can be used if the probability distribution is used as the policy function [15]. If the value function of action is used to adjust the policy directly to ensure that the choice of action is unique in the same state, the deep deterministic policy gradient (DDPG) is obtained [35,36]. The asynchronous execution of multiple simulation processes by using the multi-threading function of CPU can effectively break the correlation of training samples and obtain the Asynchronous Advantage act-critic framework

(A3C) [37]. According to the KL divergence between the old and the new policies, the learning steps are adjusted adaptively to ensure a stable policy optimization process, and then the Trust Region Policy Optimization (TRPO) algorithm can be obtained [38]. By using the proximal optimization algorithm based on the first-order optimization, the PPO (Proximal Policy Optimization) algorithm with greater stability can be obtained [39].

### 3. PDRR—A Framework for Recognizing Cyber Defense Decision-Making

The type of task determines how reinforcement learning algorithms are used. In an intranet network, the attacker plans to explore the network environment and get closer to the target node, who hopes to avoid being detected by defenders. Faced with the "ghostly" attacker, the defender should make good use of the defend resource to prepare for all possible situations. These different situations will determine the suitability of decision-making models and reinforcement learning algorithms. An example of the internal attack–defense is shown in Figure 3.

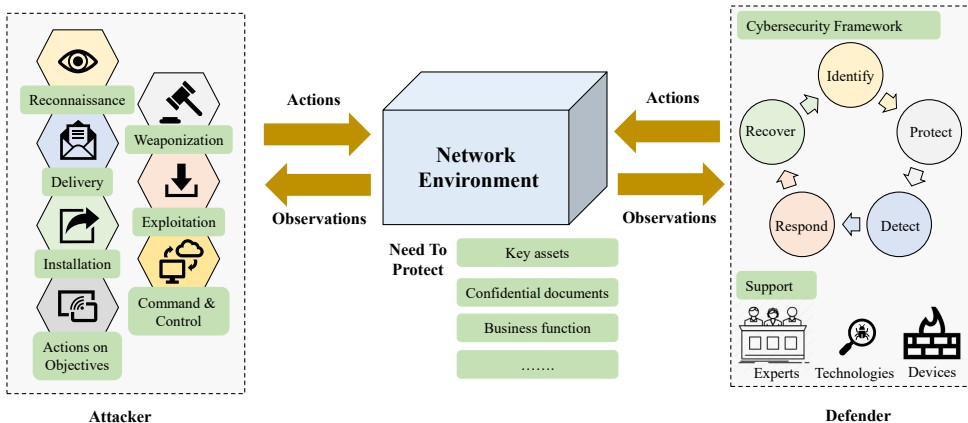

**Figure 3.** An example of the internal attack–defense.

Therefore, we propose the PDRR framework to distinguish different cyber defense decision-making tasks. The PDDR framework dynamic make decisions over the life cycle of threats, and integrates the perspectives of both attack and defense, which includes four modules, as shown in Figure 4.

**Pentest:** Pentest, also known as penetration test, attack planning, risk assessment, or vulnerability assessment, is a method to identify potential risks by simulating the decision-making process of attackers. The purpose of the test is not only to find a single vulnerability in the target network, but to form a series of attack chains (multi-step attacks) to reach the target node. The path from the starting node to the target node is called the attack path.

**Design:** Design is the decision-making task made by defenders to optimize the deployment of defense resources and design defense mechanisms before cyber attacks occur. The defense resources could be the experts, the number of security devices or the deployment location.

**Response:** Response in the process requires defenders to be able to identify, analyze, and deal with threats. In the real world, defenders often need to analyze a great number of real-time traffic data to prevent or detect threats.

**Recovery:** Recovery after the event refers to the decision-making mechanism that takes effect after the attack has occurred, to restore business function or minimize loss as soon as possible.

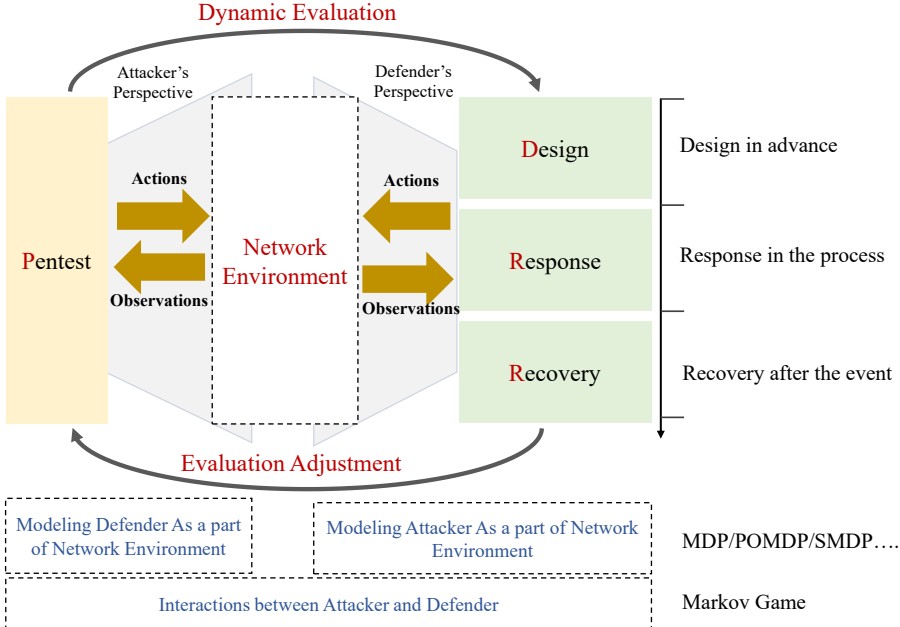

**Figure 4.** The PDRR framework.

The PDRR framework describes cyber defense decision-making at different stages of threats in an interactive form, forming a dynamic cyclic process. The "Pentest" task identify network defense from an attacker's perspective, dynamically evaluating potential risks by simulating the attacker's behaviors, where the attacker's observations include connectable hosts, exploitable vulnerabilities, currently privileged nodes and so on, and the attacker's actions are usually to obtain network informations and attack the selected nodes based on these informations, such as exploiting vulnerabilities, using passwords to log in to the host and so on. The "Design", "Response" and "Recovery" tasks, recognize the cyber defense at stages from the defender's perspective. Take the "Response" task for example, the defender's observations include detecting whether hosts in the network are attacked, whether there are abnormal behaviors, and whether vulnerabilities are patched and so on. While the defender's actions include reimaging hosts, patching vulnerabilities, and blocking subnets.

In addition to understanding cyber defense decisions as a whole, the biggest benefit of this is that the appropriate decision model can be selected based on the task. As the purpose of cyber defense decision-making at various stages is different, some factors can be selectively ignored or weakened when the conditions are not met. For example, when evaluating a Design module, the consequence is needed rather than the process. Depending on whether the adversary or the environment is considered, single-agent or multi-agent reinforcement learning algorithms can be selected. In terms of single agent's decision-making, the attacker or the defender shall be considered as part of dynamic network environment. Considering that attack and defense agents can interact according to certain strategies, the model should be constructed as a Markov Game.

## 4. Survey from the Attacker's Perspective

This section focuses on sorting out representative literature about the "Pentest" task, which is from the attacker's perspective. First, problem characteristics are discussed and, secondly, we summarized and compared the problem characteristics, decision-making model, policy and description of related research.

### 4.1. Problem Characteristics

In a "Pentest" task, we identify possible risks from the attacker's perspective. The attacker needs to find a feasible attack path to the target node in an unknown dynamic

network environment, its objective function is usually to find the random shortest path to the target node, which is the default objective function of this section. However, in order to verify the performance of the algorithm, different objective functions are also designed.

Dynamic network environment is composed of static configurations and dynamic behaviors. Static configurations refer to network topology, installed softwares, vulnerabilities, accounts and so on. Dynamic behaviors mainly refer to behaviors of three different roles, the attacker, defender and normal user. These different behaviors can change the state of the environment, which bring uncertainty to the decision-making process [40]. In this paper, the uncertainty in the task can be divided into four parts.

**Unknown environment:** Due to the fact that the attack process takes place in a non-cooperative network environment, the agent needs to obtain information (such as to find the host can be connected, to figure out whether the vulnerability can be exploited and so on) by exploring the environment. Based on the information, the agent chooses appropriate actions to complete the exploration of the environment.

**Partial observability:** Since the attacker agent is in a non-cooperative environment, after the agent taking actions, it often cannot judge whether received observations are real states of the system, and can only maintain a certain degree of confidence. In this case, the attacker needs to constantly update its belief in the state of the environment based on the obtained information, and then select appropriate actions.

**Competitive environment:** Security devices and security experts are the biggest hindrances to the intrusion process. In order to clear the foothold to prevent critical systems from being controlled by the attacker, the defender take countermeasures include restarting computers, shutting down hosts, installing patches, and blocking subnets. Consequently, this will affect the decision-making process of the attacker.

**Non-competitive environment:** Normal users play a third role in a network environment besides the attacker and the defender. They may perform operations such as restarting hosts, establishing connections between two hosts, and installing new softwares. These actions still cause uncertainty in the decision-making process.

*4.2. Related Work*

When the attacker treats the defense and ordinary users as part of the environment, the attacker decision-making model can be modeled by MDP, POMDP and other models. If the attack agent and the defense agent are regarded as competitors, the Markov Game can be used to model the interaction between them. In early stage, the "Pentest" is mainly solved by the planning-based method. With continuous and deep research on the reinforcement learning method, more and more researchers use reinforcement learning to solve this problem. Table 1 summarizes representative research work in this direction in recent years.

**Table 1.** Survey from the attacker's perspective.

| Type | Ref | U | P | C | N | Policy | Description |
|---|---|---|---|---|---|---|---|
| POMDP | [41] | ✓ | ✓ | × | × | SARSOP [42] | (1) Introduce beliefs to reflect uncertainty; (2) Excessive computational complexity. |
| | [43] | ✓ | ✓ | × | × | Improved SARSOP | (1) Introduce network structure to reduce complexity; (2) Not applicable when structure changes. |
| | [44] | ✓ | × | × | × | Improved RL | Introduce the network information gain to evaluate actions. |
| | [45] | ✓ | ✓ | ✓ | × | Improved Bayesian RL | Introduce the information decay factor to reflect the influence of the adversary. |
| | [46] | ✓ | ✓ | × | × | Improved GIP [47] | (1) Path coverage exceeds that of human in small networks; (2) Computational cost exceeds that of human on medium LANs. |
| MDP | [40] | ✓ | × | × | × | - | Modeling as deterministic planning. |
| | [48] | ✓ | × | × | × | Q-learning, Deep Q-learning | (1) Build a system to simulate attack; (2) Ignore the defender and normal user. |
| | [49] | ✓ | × | × | × | DQN with WA [50] | (1) Introduce "WA" to reduce action space; (2) Use graph embedding method to abstract state; (3) Low-dimensional vectors lacks interpretability. |
| | [51] | ✓ | × | × | × | RNN | The RNN algorithm can also work well when the environment changes. |
| | [52] | ✓ | × | × | × | A2C, Q-Learning, SARSA | (1) The A2C algorithm performs better than other algorithms. (2) State representation lacks interpretability. |
| | [53] | ✓ | × | × | × | DQN, PPO | (1) The stability and generalization are disscussed for the first time. (2) Current algorithms have poor generalization performance. |
| Markov Game | [54] | ✓ | × | ✓ | ✓ | Planning-based | Design dynamic network configurations to compare attack and defense interactions. |
| | [55] | ✓ | × | ✓ | × | Strategy sets | (1) The adaptability of algorithms to the adversary is verified. (2) The task is far from real world. |
| | [56] | ✓ | × | ✓ | ✓ | Q-learning, DQN, XCS [57] | (1) Designed a attack-defense multi-agent system. (2) Different action usages are not discussed. |
| | [58] | ✓ | × | ✓ | × | DQN | (1) Verify the algorithm's applicability. (2) The defined model is difficult to scale. |

Note: U indicates "Unknown environment", P indicates "Partial observability", C indicates "Competitive environment" and N indicates "Non-competitive environment".

**Pentest based on POMDP:** The POMDP uses the "belief" to reflect uncertainty in attack decision-making. Sarraute et al. [41] first introduce POMDP to simulate "Pentest" task, the goal of the task is to find the shortest path to the target node. Due to the high complexity of POMDP algorithms, algorithms can only work in the environment of two hosts. In order to reduce the computational complexity, Sarraute et al. use decomposition and approximation to abstract network attack at four different levels [43]. The algorithm decomposes the network into a tree of biconnected components, then gradually calculates the cost of attacking subnets in each component, hosts in each subnet, and each individual host. Although this method reduces complexity, it ignores the consequences of possible changes in the network environment.

Due to the limited complexity of planning algorithms, some researchers began to use reinforcement learning algorithms to solve this problem. Zhou et al. introduce the network information gain to as the signal for evaluating actions' reward [44]. The information gain is equal to $H(P) = \sum_{j=1}^{|p|}(p_j \log p_j + (1 - p_j) \log(1 - p_j))$, where $P_j$ represents the vector of operating system, open service, port, and protection mechanism probability distribution before (after) taking actions. Experiments showed that RL algorithms with information gain are more effective than other algorithms at finding attack paths.

In addition to considering an "unknown environment", Jonathon et al. also took into account the influence of the opponent, and proposed an information decay factor to represent the observation of the attacker [59]. The author used the Bernoulli process to model the defender's behaviors, and verified that the Bayesian Reinforcement Learning (BRL) algorithms combined with the information decay factor is significantly better than the baseline algorithm in terms of winning rate and reward.

Ghanem built an automated penetration testing system called "IAPTS" to solve the problem of automated Pentest in large network environments [46]. The system preprocesses the original data, and obtains the evaluations by constructing POMDP model to use PERSEUS [15], GIP [47] and PEGASUS [60] algorithms in combination, and then made it available to human experts. The experiments are carried out in small and medium networks. Experimental results showed that the improved GIP algorithm performed better than manual experts on small networks, but the computational cost was higher on medium LANs, requiring more in-depth research on the algorithm.

**Pentest Based on MDP:** Through existing research, it can be seen that when using POMDP to model the attack decision-making process, due to the computational complexity of algorithms, the scale of applicable scenarios is limited. Therefore, more and more researchers choose to model the "Pentest" task with MDP, where the outcomes of actions are deterministic [40]. This change allows researchers to build the task with more consideration for the similarity of actions to the real world, the changes in the environment, and the presence of the opponent and so on.

Schwartz designed and built a new open-source network attack simulator (NAS) for studying the attack decision-making process [48], and compared different RL algorithms' convergence times. This system promoted the research of reinforcement learning algorithms in the direction of "Pentest", but the abstraction of state space is too simple and lacks consideration of the dynamic environment, so it needs to be further deepened.

Nguyen et al., focused on the problem of large-scale action space reduction [49]. The author combined the DQN algorithm with "Wolpertinger Architecture(WA)" [50] to embed multilevel action. The sensitive machine attacked proportion of the proposed algorithm was obviously better than other algorithms, and was also applicable in the medium and large-scale network. Besides, the graph embedding method is first used to represent the state space, which is worth exploring.

By focusing on how to reduce the number of features represented in the state space and how to approximate the state-action space with a small set of features, Pozdniakov et al. performed the experiment to compare the advantages and disadvantages of three algorithms: Q-learning based on table, RNN with Q-learning and RNN with Q-learning after network configuration changes [51]. The results showed that, compared with Q-learning based on table, RNN with Q-learning can converge faster and respond to the change of environment, showing the potential of model-free learning for automated pentest.

Maeda et al., used the real payloads of attack tools as actions for attack decision-making, and applied A2C, Q-learning and SARSA algorithms to solve the task [52]. The experimental results showed that the A2C algorithm has the best effect in cumulative average reward. However, the defined state representation in the article lacks a certain degree of interpretability.

In order to test the stability and generalization of various DRL algorithms [53], Sultana et al. built five experimental scenarios with different topologies, services, and application configurations based on the Network Attack Simulator (NASim) platform [59]. In the test

of stability, the returns of DQN algorithm and PPO were relatively stable. In the test of generalization, when the evaluation network deviates from the training environment, the performance of DQN and PPO algorithms failed to adapt to all experimental scenarios, even worse than the performance of random algorithm in the same environment, which pointed out the possible research directions.

**Pentest Based on Markov Game:** Markov Game in "Pentest" treats the attacker and the defender as the competing agents and designs a Game model to compare the effectiveness of the two sides's policies. Applebaum et al. systematically modeled dynamic network environment, and used the method of statistical analysis to discuss the winning and losing situations of both players under different environment configurations [54].

Elderman et al., modeled the attack–defense game as zero-sum two-player Markov Game with incomplete information. In the game, agents played in turn-based interactions based on their observations. The attacker tried to reach the target node, while the defender tried to protect the network and stop the attacker. In this process, the decision-making of both sides was influenced by the hidden information [55]. The experiment compared the winning and losing of players using different strategies, and found that neither side can ensure long-term victory under the same strategy.

Niculae built a more realistic multi-agent system, where included the interactions among attackers, defenders and normal users [56]. In the system, ten attack actions, three defense actions and three normal user actions were designed. In addition, attributes such as reliability, duration, noise, and crash probability were added to the attack action to describe the uncertainty of the interaction process. The experiments compared reinforcement learning algorithms such as DQN and Q-learning with rule-based algorithms, results showed that the DQN algorithm had stronger generalization ability, but required a large amount of training time. However, it did not analyze how often the different types of actions designed were used in its algorithm.

Bland et al., used the Petri net as a modeling tool to simulate the attack decision-making process [58]. The author designed several groups of comparative experiments in which static attackers versus dynamic defenders and dynamic attackers versus dynamic defenders. In these comparative experiments, reinforcement learning algorithms using greedy policies improved performance over time, both in achieving player goals and reducing costs. However, using the Petri net requires a detailed definition of the transition between different states, which makes the work less applicable.

## 5. Survey from the Defender's Perspective

Unlike research from the attacker's perspective, the studies from the defender's perspective needs to consider various situations in the defense decision-making. Based on the PDRR framework, this section reviews research in terms of the three defense phases: Design, Response, and Recovery.

### 5.1. Problem Characteristics

Although the defender can control the internal network, the agent does not know whether the attack exists and the possible consequences of the attack. Therefore, its decision-making process still has the following challenges.

**False Positive Rate:** In the abnormal detection process, the detection by the defender will cause a certain false positive rate due to the existence of the anti-detection technology of unknown and known threat samples. This will happen for the high degree of fitting between the existing samples and the malicious samples, or forged signals sent by the attacker to the defense detection algorithm. The existence of a false positive rate will further improve the complexity of learning, leading to the wrong judgment of defense decision.

**Local Information Known:** Security experts find it difficult to grasp all the information of the network environment in real time due to the dynamic and huge scale of the network environment, though they have the authority to manage the network environment. Therefore, the defender can only manage and maintain the network environment of limited

scale, and such local limitation can be alleviated by means of multi-agent cooperative communication, while increasing the difficulty of decision-making.

**Constraint of Resources:** The attack may be launched at any time using different vulnerable points, and the protection, detection, response and recovery of defense will take up or consume certain resources. Given this, an unrestricted commitment of defense resources is impossible. These resources may be the cost of equipment deployment, or personnel, time and other cost factors. Hence, in a network defense scenario, the defender needs to respond as quickly as possible and cut the cost as much as possible to complete the task.

Table 2 summarizes and compares representative research literature from the defender's perspective. Like the attacker perspective, we also classify these three tasks from the adopted decision-making model, and compared decision-making models, task, strategies used and descriptions of related research.

**Table 2.** Survey from the defender's perspective.

| Type | Model | Ref | F | L | C | Policy | Description |
|---|---|---|---|---|---|---|---|
| Design | MDP | [61] | × | × | ✓ | Value Iteration | Dynamic optimization algorithm and manual intervention are combined. |
| | | [62] | ✓ | × | ✓ | Q-learning | Proposed a new adaptive control framework. |
| | | [63] | × | ✓ | ✓ | Q-learning | Proposed a spoofing resource deploy algorithm without strict constraints on attacker. |
| | Markov Game | [64] | – | × | ✓ | Q-learning | An algorithm that can obtain the optimal security configuration in multiple scenarios; |
| Response | MDP | [65] | ✓ | × | ✓ | DDQN, A3C | (1) Design an autonomous defense scenarios and interfere with training process; (2) observation is less limited. |
| | | [66] | ✓ | ✓ | × | DQN | (1) The convergence is accelerated by expert artificial reward (2) Limited reward setting. |
| | | [67] | × | × | × | Q-learning | (1) Knowledge graph is used to guide the algorithm design (2) High trust in open source data. |
| | | [68] | – | × | ✓ | Value Iteration | (1) Introduces a botnet detector based on RL; (2) Action cost not taken into account. |
| | | [69] | – | × | ✓ | Reward structure | Multi-agent network with collaborative framework and group decision-making. |
| | | [70] | × | × | × | Sarsa | (1) Hierarchical collaborative team learning can extend to large scenarios; (2) Difficult to guarantee convergence. |
| | Markov Game | [71] | × | ✓ | ✓ | Q-learning | (1) Pareto optimization is used to improve Q-learning's efficiency; (2) The attacker takes random actions. |
| | | [72] | × | ✓ | ✓ | Q-learning | (1) A defense policy selection simulator is constructed; (2) Random attack actions. |
| | | [73] | ✓ | ✓ | × | Q-learning | (1) Respond to attacks with limited information; (2) Rely on expertise to evaluate rewards. |
| | | [74] | ✓ | ✓ | ✓ | MA-ARNE | Solving resource utilization and effective detection of APT with MARL. |
| | POMDP | [75] | × | ✓ | ✓ | Q-learning | POMDP is modeled based on Bayes attack graph. |
| | | [76] | × | ✓ | ✓ | Q-learning | (1) The transfer probability is estimated using Thompson sampling; (2) Fixed attack transfer probability. |
| | | [77] | ✓ | ✓ | × | Value Iteration | (1) Adaptively adjust the recognition configuration according to the false alarm rate; (2) Cannot face adaptive attackers well. |
| | | [78] | ✓ | ✓ | ✓ | Q-learning | (1) Multi-agent collaboration model based on signal and hierarchy; (2) Cannot guarantee the convergence. |
| | | [79] | ✓ | ✓ | ✓ | Q-learning | (1) A decentralized threat perception model is proposed; (2) Value functions needs expert knowledge. |
| | SMDP | [80,81] | – | × | ✓ | Q-learning; | The interaction in the honeypot is modeled as a semi-Markov process. |
| Recovery | MDP | [82] | – | × | × | DDPG | (1) The continuous numerical method is used to obtain a better real-time recovery policy; (2) Low convergence rate. |
| | | [83] | – | × | × | Q-learning | (1) Proposed an effective strategy based on a realistic power flow model with RL; (2) Low convergence rate. |
| | | [84] | – | × | × | DDPG | Applied reinforcement learning to backup policy. |

Note: F indicates "False Positive Rate", O indicates "Local Information Known ", C indicates "Constraint of Resources".

### 5.2. Design

"Design" task refers to deploying resources or designing mechanisms to deal with different cyber attacks.

The moving target defense (MTD) is proposed to "change the rules of the game" facing with attacks [85]. MTD mainly works to confuse the attacker by constantly shifting the attack surface, increase the attack cost and difficulty, and improve the resilience of the defense system. Ganesan et al. [61] proposed a dynamic stochastic model for scheduling security analysts. The risk can be kept below target levels by coordinating the allocation of staff and sensor resources.

Winterrose et al. [62] has developed a method based on online reinforcement learning to achieve the ideal balance between system security and system performance in a dynamic threat environment. With this method, the best parameters needed to defend against adversaries in the security performance space can be autonomously computed even as the threat changes. However, it usually consumes a lot of energy and resources.

In addition to changing the attack surface, researchers have also countered the attack by deploying network deception devices such as honeypots. Wang et al. [63] used Q-learning-based agents to find the best policy of deploying spoofed resources. The algorithm solves the problem that the static deployment of spoofing resources is easy to identify and bypass. In the actual network experiment, this method can reach the defense success rate of 80%.

Jiang et al. [86] proposed the concept of "moderate security", which means that we should seek a risk-input balance when considering the constraints of resources and other actual conditions. In other words, the limited resources should be used to make a reasonable decision, so the game theory can be used to study the strategies of both players.

A good defense mechanism "Design" should also be found in attack–defense confrontation. Panfili [64] et al. obtained the optimal security configuration by searching Nash Equilibrium of multi-agent attack and defense game. It can minimize the damage caused by an attack, even making it less than the cost of executing the attack policy.

### 5.3. Response

"Response" task requires the defender to be able to detect abnormal network traffic and take appropriate response measures, especially in response to unknown attacks.

A zero-day vulnerability is a vulnerability that has not been disclosed. Before releasing system patches for zero-day vulnerabilities, how to actively establish a response mechanism during the repair window to deal with unknown cyber attacks is a challenging problem. Sun et al. [71] modeled the network attack and defense as a zero-sum multi-objective game in a simple network topology, and proposed the Q-learning algorithm with pareto optimization for the "Response" task, where helps network security analysis improved significantly. Based on the above research, Sun et al. improved the granularity and authenticity of the network model [72]. A set of defense policy selection simulator was built to the Q-learning algorithm with pareto optimization. However, in the above two studies, the attackers used random actions that did not fit well with real attacks.

Hu et al. [75] modeled the "Response" task as a partially observable Markov decision process on a bayesian attack graph. In the experiments, Q-learning was used to identify cyber attacks, and verified the performance of the algorithm in a small network. However, it is assumed here that the process of responding is phased, making it difficult to detect online. Based on the previous work, Hu et al. proposed a new adaptive network defense mechanism [76]. The task was modeled as a POMDP problem with uncertain state transition, and the state transition probability was estimated by using Thompson sampling. The optimal defense action was obtained based on reinforcement learning without any false positives. Based on the real network attack numerical simulation, the cost-effectiveness of the algorithm was verified. However in the real world, the attack value may change continuously.

Han et al. [65] explored the feasibility of reinforcement learning algorithm for autonomous defense in SDN and the false positive problem of attack. In their experiment, the defense agent used various reinforcement learning methods to learn the best operations to protect critical servers while isolating as few nodes as possible. In addition, in their adversarial training, the attacker can damage the reward and state of the defensive agent, thus causing false positives. Therefore, the reinforcement learning may make sub-optimal or even wrong decisions, which indirectly proves that the adversarial training can reduce such negative effects for the defender.

The false positive rate and defense expenses will increase with the rising number, type and complexity of network attacks. As an attacker will often hide from the network's constant real-time monitoring after successfully entering a network environment, the defender should implement appropriate means to identify and respond to abnormalities in time, formulate corresponding response mechanisms to handle abnormal detection results containing false positive rate. Chung et al. [73] proposed an automatic detection technique to respond to the hostile behavior of suspicious users. This is based on the game model of expert knowledge and uses Q-learning to solve. Experiments based on simulation showed that, when the information of the opponent is limited, the Naive Q-learning can effectively learn the behavior of the opponent and has better performance than other algorithms with limited assumptions. Due to the lack of consistency in security metrics, the damage and reward of attacks were mainly based on the expert knowledge.

Liu [66] studied the interactive reinforcement learning method which was used to improve the adaptability and real-time performance of intrusion detection. Based on the experience replay buffer and Long Short-Term Memory (LSTM) optimization learning, the feedback of network security experts was added in the reward to accelerate convergence. In addition, the expert also reported different rewards based on the difference between true positive and false negative rates. An excellent detection effect was achieved on the NSL-KDD dataset [87].

Sahabandu et al. [74] studied how to weight the resource efficiency and detection effectiveness in dynamic information flow tracking analysis models. Their game model captured the security costs, false positives and missing positives associated with dynamic information flow tracking, and a multi-agent average reward Nash equilibrium algorithm is proposed (MA-ARNE) was proposed, which was able to converge to an average reward Nash equilibrium on the ransomware data set.

Piplai et al. [67] used the prior knowledge represented by the network security knowledge graph to guide the reinforcement learning algorithm to detect the malicious process, and applied the prior knowledge mined from the open text information source describing the malicious network attack to guide the reinforcement learning algorithm to adaptively change parameters to adjust the value function and explore the probability. The simulation experiment proved that such system was better than the basic reinforcement learning system in terms of detecting malicious software.

Modern botnets often operate in a secretive manner, which make it difficult to detect. Venkatesan et al. [68] based on reinforcement learning, detected the presence of botnets by deploying honeypots and detectors in a simulated environment, and performed the comparative experiments with static policies in PeerSim [88]. The results showed that the performance was improved significantly.

Alauthman et al. [77] proposed a detection system combined with reinforcement learning technology, in which, the network traffic reduction technology was mainly used to deal with large-scale network traffic. As a result, the botnet hosts with high accuracy (98.3%) and relatively low false positive rate (0.012%) can be detected, which was better than other detection methods. Dynamic improving system based on reinforcement learning can alleviate the dynamic changes existing in botnets, but it cannot well face the methods of avoiding detection taken by botnet administrators, such as rootkits (A malicious software that can hide itself and files and networks from the host).

In response to different attacks, Honeypot technology is adopted to protect the network. Honeypot technology is an active defense technology which can effectively defend network resources by deceiving attackers to obtain information [89]. Huang et al. [80,81] applied a semi-Markov decision process to characterize the random transitions and linger time of the attackers in the honeypot, and designed a defense scheme where the agent actively interacted with the attacker in the honeypot to increase the attacker's attack cost and collect threat information. According to the numerical results, the proposed adaptive policy can quickly attract attackers to the target honeypot and interact with them for a long enough time to obtain valuable threat information. At the same time, the penetration probability was kept at a low level.

The multi-agent system, a classic model of distributed artificial intelligence, can provide better adaptability and more effective decision-making for cyber defense [90]. Miller and Inoue [69] used the Synergistic and Perceptual Intrusion Detection with reinforcement (SPIDeR) [91] and an agent with its Self-Organizing Map (SOM) to cooperate to detect anomalies. Additionally, decisions were made by the central blackboard system in SPIDeR based on reinforcement learning. SPIDeR showed positive results in 1999 KDD Cup [92].

Malialis [70] also introduced a multi-agent cooperative architecture to build a layered anomaly detection framework, so as to improve the response speed and scalability of intrusion detection, and studied the cooperative team learning of agents. However, similar to the study of Servin et al. [78], the large-scale collaborative learning of distributed agents was difficult to guarantee the convergence.

Liu et al. [79] designed a collaborative network threat perception model (DDI-MDPS) based on decentralized coordination to solve the problems such as high pressure, low fault tolerance, low damage resistance and high construction cost of the static centralized network intrusion detection system (NIDS). In addition, they tested the DDI MDPs model based on the open data CICIDS2017, and the simulation results proved that the interaction between multiple agents enhanced the network threat perception. However, designing value functions for agents to deal with the network threat perception problems in unknown networks largely relies on prior domain knowledge.

*5.4. Recovery*

"Response" task needs the defender to be able to take measures to restore network functionality after cyber attacks have occurred.

In the case of unavoidable exceptions or damages, the defender should take adequate measures to restore the normal operation of the network in time and minimize the impact caused by attacks. For critical power infrastructure, the system can quickly recover by using reinforcement learning technology after detecting and removing the invasion [93]. A malicious attacker can trip all transmission lines to block power transmission as long as it takes control of a substation. As a result, asynchrony will emerge between the separated regions interconnected by these transmission lines.

Wei et al. [82], proposed a recovery policy for how to properly choose the reclosing time, which uses a deep reinforcement learning framework, so as to reclose the tripped transmission lines at the optimal reclosing time. This policy was given with adaptability and real-time decision-making ability to uncertain network attack scenarios. Numerical results showed that the policy proposed can greatly minimize the impact of network attacks in different scenarios.

Wu et al. [83] modeled the cascade failure and recovery process in the power system as MDP and used Q-learning to identify the optimal line sequence recovery sequence. The recovery performance of this method was proven to be better than that of the baseline heuristic recovery policy in IEEE57 simulation systems.

Debus et al. modeled the threat model and decision-making problem mathematically for the traditional network defense methods, and transformed the problem of finding the best backup policy into a reinforcement learning problem [84]. The storage device updating scheme based on the proposed algorithm can achieve or even exceed the performance of

the existing scheme. In addition, thanks to the function approximation property of the policy network, the algorithm not only provides a fixed set of update rules and steps, but also can deal with the deviation from the planned scheme and dynamically adjust the backup scheme back to the best route.

## 6. Open Directions and Challenges

According to existing literature, It can be seen that most of the work still remains on how to apply reinforcement learning algorithms. Therefore, this paper makes an outlook from both reinforcement learning and cyber defense decision-making perspectives.

### 6.1. Challenges in Reinforcement Learning Algorithms

In terms of existing studies on reinforcement learning, reinforcement learning algorithms are too simple to solve the real problems in cyber defense decision-making. Therefore, the reinforcement learning research in network defense decision-making still needs to overcome the following challenges:

**Appropriate Objective Function and Optimization Method:** Different from supervised learning, there is no unified optimal goal because the reinforcement learning has no labels. Objective functions are designed in terms of stability, convergence rate and accuracy of value function. However, what kind of objective function is more suitable for the policy solution of network attack and defense decision-making task needs to be further studied. In addition, the optimization methods such as batch gradient descent, random gradient descent, least squares, Kalman filtering, Gauss process optimization, proximal optimization, Newton method etc. should be further discussed given the objective function.

**Value Functions and Policy Gradients:** Both reinforcement learning algorithms are based on the value function and policy gradient ascending algorithm which directly defines a policy exist in many network attack and defense decision-making tasks. Reinforcement learning algorithms based on a value function are usually used in discrete action space, while policy gradient based reinforcement learning algorithms are usually used in continuous action space. However, how to define a policy for a specific task should be considered carefully [15].

**Sparse Reward Challenge:** In network attack and defense decision-making tasks, rewards are usually sparse. Sparse reward is one of the important factors affecting the convergence rate of reinforcement learning. Using reward shaping to make sparse rewards denser can effectively improve the convergence rate of learning, one of whose cores is to satisfy policy invariance [94]. There is another scheme, where the decision-making behavioral data of experts are collected and the techniques such as inverse reinforcement learning are used to reverse the reward function [95,96].

**Non-stationary Challenges:** Reinforcement learning assumes that the environment is stationary: Time-independent uncertainty. However, in the network attack and defense game, the following two situations lead to nonstationary environment:

(1) Environmental uncertainty may be time-varying [97], for example, the network topology disturbed by uncertain factors (abnormal shutdown, etc.) may lead to the failure of nodes;

(2) The competitive game between attacking and defending agents causes the environment to respond to different action pairs.

Such a nonstationary environment will seriously affect the convergence of reinforcement learning algorithm. Therefore, it is necessary to conduct context detection, track potential changes [98], predict [99], and construct a model to accommodate it, such as meta-reinforcement learning [100] and worst case idea [101] etc.

**Non-Ergodicity Challenges:** Reinforcement learning assumes that the environment satisfies Ergodicity, also known as Ergodicity of each state, which means that the probability of each state is non-zero and converges to a definite value. However, in the network attack and defense game, this ergodicity is broken, causing that some nodes in the topology are broken at a very low possibility. This brings a strong challenge to reinforcement

learning. Given this, how to design a stable and efficient reinforcement learning under the assumption of non-ergodicity is much needed, such as using utility function based on a value function [102–104].

### 6.2. Challenges in Cyber Defense Decision-Making

Network defense decision-making requires in-depth study of problem characteristics in combination with real scenarios, so as to provide a basis for more convenient simulation experiments and theoretical analysis, as well as better evaluation of defense decision-making effects. The challenges in network defense decision-making are as follows.

**Task model:** In order to accurately characterize the characteristics of cyber attack and defense interactions, the researchers proposed FilpIt [105,106], Hide-and- Seek [107–110], two-person zero-sum game [55] and other task models to analyze the decision-making behavior of both attacker and defender. The above models have abstracted the characteristics of network attack and defense, such as concealability and dynamics, but it is still difficult to provide guidance for the decision-making process due to the theoretical reasons as follows: (1) The design of environment, action and state of the existing models is abstract and lacks the mapping with the real world. (2) Most network attack and defense games are abstracted as two-person zero-sum games, whose design of reward and ending mechanism cannot accurately reflect the decision-making preferences of both attacker and defender. Therefore, to conduct a more in-depth analysis on the characteristics of attack and defense game in actual task scenarios such as Pentesting and virus infection is necessary.

**Table 3.** Comparison of Current Network Training and Testing Platform.

| Network Implementation | Model | Literature | Convenience | Fidelity | Expansibility |
|---|---|---|---|---|---|
| Network Simulation | MulVAL [111] | [112–115] | Middle | Low | Low |
| | Petri net [116] | [58] | Low | High | High |
| | NS3 [117] | [118] | Middle | High | Middle |
| | Mininet [119] | [65] | Middle | Middle | Middle |
| Testing platform | NASim [59] | [48] | High | Middle | High |
| | CyberBattleSim [120] | - | High | Middle | High |

**Simulation platform:** Previous studies mainly verify the effectiveness of network attack and defense decision-making algorithm by MulVAL [111], NS3 [117], Petri net [116] real data playback or using virtual machine network. In Table 3, the relevant work of the existing training and testing platform is summarized, which is divided into three indexes of high, middle and low to measure the convenience, fidelity and expansibility of each network training and testing platform.

(1) Convenience: How easy the platform is to use, which is evaluated on the basis of the size of the package, the platform on which it is running, the language in which it is compiled, and how it is installed.

(2) Fidelity: Closeness to the real world, which is mainly evaluated from action design, network environment, evaluation indexes, etc.

(3) Expansibility: The difficulty of secondary development, which is evaluated based on the open source degree of the platform and the difficulty of the experiment of researchers.

Obviously, using NS3, real data playback and other methods to build the training test environment will bring a lot of unnecessary details in terms of decision-making, such as raw data preprocessing or traffic packet parsing. As a result, the processing complexity of the algorithm is increased. However, Petri net or MulVAL and other methods to build the training testing environment need a lot of pre-defined work, which is difficult to reflect the dynamic of network attack and defense. Therefore, it has become a current development trend for researchers to use multi-agent methods to build training and testing platforms for attack and defense, such as the Network Attack Simulator [59] by Schwartz and Cyber-BattleSim [120] by Microsoft. However, these training test platforms are still developing,

which are difficult to define and expand the elements of attacking and defending actions, states and tasks flexibly. So, to further develop diversified training environments to meet the needs of decision-making algorithm is required.

**Best response strategy:** In single-agent reinforcement learning, an agent repeatedly interacts with the environment to optimize its policies until there is no way to provide further performance (as measured by rewards or other metrics). However, the attacker in the process of defense may have more than one policy; the defender therefore shall comprehensively consider different policies, different starting points, and even different attackers with different capabilities, so as to make the optimal response as far as possible. In this case, empirical game analysis and reinforcement learning [121] shall be used to find the optimal defense response policy in the policy space.

**State representation:** The state of the decision-making process needs to consider elements such as network topology, vulnerabilities, and accounts, and Network topology is a natural graph structure. Therefore, graph neural networks, graph convolutional deep neural networks [122,123] and graph deep neural networks [124,125] can be used to effectively represent the state and observation of network attack and defense decision-making processes. However, only little literature [50] discusses the use of graph embedding method, more in-depth research should be performed on how to better represent the multidimensional characteristics in the attack and defense decision-making and analyze the differences between different representation methods.

**Centralized control and distributed control:** In the existing research on network attack and defense decision-making, most of them adopt distributed cooperative mode to control multiple agents for intrusion detection and anomaly detection. However, large-scale collaborative learning of distributed agents was difficult to guarantee the convergence [69,70,78]. Furthermore, the hostile behavior has gradually taken on the purposeful, organizational characteristics in the current network attack and defense decision-making environment. Therefore, the most important work in the next stage is to study a set of centralized control learning methods, and multiple agents are controlled to make unified decisions at the same time, in order to respond to the hostile behavior with clear goals and organizational discipline.

## 7. Conclusions

Internet technology is the foundation of cyberspace, while network security is an inevitable problem in the development of cyberspace. Reinforcement learning, as a new technology, has been used by many researchers to solve the cyber defense decision-making problem. In this paper, based on the summary of intelligent decision-making models and reinforcement learning technology, representative literature is summarized and contrasted from perspectives of the attacker and the defender, respectively. The former focuses on the uncertain process of attack decision-making, and the latter focuses on addressing the various scenarios that cyber defenses may face.

This paper is prepared to sort out the technology and application of reinforcement learning in the field of network attack and defense decision-making, and provide ideas for further research on network attack and defense decision-making by systematically analyzing the advantages and disadvantages of existing research. At the end of the paper, the challenges of reinforcement learning and network defense decision-making are summarized. It can be seen that reinforcement learning shows the potential to solve cyber defense decision-making problems, but the task models, platforms and algorithms still need further research to promote this direction.

**Author Contributions:** Conceptualization, W.W.; investigation, D.S. and F.J.; methodology, W.W.; resources, X.C.; writing—original draft, W.W.; writing—review & editing, C.Z. All authors have read and agreed to the published version of the manuscript.

**Funding:** Supported by National Natural Science Foundation of China, General Program, 71571186.

**Institutional Review Board Statement:** Not applicable.

**Informed Consent Statement:** Not applicable.

**Data Availability Statement:** Not applicable.

**Conflicts of Interest:** The authors declare no conflict of interest.

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
