# Peer review of "Research and Challenges of Reinforcement Learning in Cyber Defense Decision-Making for Intranet Security"

_algorithms, doi:10.3390/a15040134_

Round 1

Reviewer 1 Report

The paper presents a survey about using reinforcement learning approaches in Cyber-Security. In particular, the authors firstly propose a framework named PDRR, which clearly separates the different cyber defence decision making tasks. Then, a systematic literature review is illustrated according to various aspects highlighted by the framework.

The topic of the paper is quite critical and up to date. Based on a schematisation of the entire cyber-defence process, the authors provide a clear vision of several aspects that have to be considered. For each of them, they have analysed related literature studies.

The analysis results provide an overview of current challenges arising for the RL algorithms and cyber-defence decision-making. Specifically, the analysis results highlight the open issues to be further investigated and give helpful research directions to overcome the current limits of existing approaches.

The literature analysis is well described, and each aspect to be investigated is clearly identified. I think some issues arising from the literature about RL in the cyber-security field are also critical in other contexts where RL is exploited. Hence, I believe that the paper indeed provides research directions in the field of cybersecurity but also generally for RL approaches.

Minor comments:

Line 151: lacks the reference

Review the text for minor typos

Author Response

Dear Reviewer:

Thank you kindly for this acknowledgement and recognition of my work. In response to the error you pointed out, we have carefully checked the errors in the references of the article and made corrections.

We really appreciate your comments and suggestions! Everything goes well!!

Your kind considerations will be greatly appreciated.

With best regards,

Sincerely Yours,

Wenhao Wang ([email protected]), Cheng Zhu ([email protected])

Reviewer 2 Report

The paper review application of reinforcement learning in cyber
defense decision-making for intranet security. The manuscript is interesting. There are just some minor comments before acceptance, as follows.

- The authors should add a block diagram to summarize the major contents presented in this study. Currently, it is a bit difficult to follow the manuscript.
- In section 3 "PDRR-A framework for identifying cyber defense decision-making". if possible, the authors should discuss about observations, and actions from both attacker's perspective and defender's perspective, as shown in figure 3?

Author Response

Dear Reviewer:

Thank you kindly for this acknowledgement and recognition of my work.

Question 1: The authors should add a block diagram to summarize the major contents presented in this study. Currently, it is a bit difficult to follow the manuscript.

Response: Thank you for your valuable suggestion. We have added a figure to describe the structure of this article, so that readers can understand this article more clearly, as shown in figure below.

Question 2: In section 3 "PDRR-A framework for identifying cyber defense decision-making". if possible, the authors should discuss about observations, and actions from both attacker's perspective and defender's perspective, as shown in figure 3?

Response: Thank you for your valuable suggestion. We describe the interactions between the attacker and the defender from a reinforcement learning perspective in Figures 3 and 4 in Section III, and further explain the "Actions" and " Observations" in different tasks. The amendments are listed as follows between line 223 and line 235.

The PDRR framework describes cyber defense decision-making at different stages of threats in an interactive form, forming a dynamic cyclic process. The ``Pentest'' task identify network defense from an attacker's perspective, dynamic evaluating potential risks by simulating the attacker's behaviors, where the attacker's observations include connectable hosts, exploitable vulnerabilities, currently privileged nodes and so on, and the attacker's actions are usually to obtain network information and attack the selected nodes based on these information, such as exploiting vulnerabilities, using passwords to log in to the host and so on. The ``Design'', ``Response'' and ``Recovery'' tasks, recognize the cyber defense at stages from the defender's perspective. Take the ``Response'' task for example, the defender's observations include detecting whether hosts in the network are attacked, whether there are abnormal behaviors, and whether vulnerabilities are patched and so on. While the defender's actions include reimaging hosts, patching vulnerabilities, and blocking subnets.

We really appreciate your comments and suggestions! Everything goes well!!

Your kind considerations will be greatly appreciated.

With best regards,

Sincerely Yours,

Wenhao Wang ([email protected]), Cheng Zhu ([email protected])

Dear Reviewer:

Thank you kindly for this acknowledgement and recognition of my work. In response to the error you pointed out, we have carefully checked the errors in the references of the article and made corrections.

We really appreciate your comments and suggestions! Everything goes well!!

Your kind considerations will be greatly appreciated.

With best regards,

Sincerely Yours,

Wenhao Wang ([email protected]), Cheng Zhu ([email protected])
